# Effectiveness of LMS Digital Tools Used by the Academics to Foster Students' Engagement

**Sibongile Simelane-Mnisi**

Higher Education Development and Support, Curriculum Development and Support: eLearning, Tshwane University of Technology, Pretoria 0183, South Africa; simelanes@tut.ac.za

**Abstract:** The purpose of this study was to examine the effectiveness of LMS digital tools used by academics to foster student engagement at the University of Technology in South Africa. International studies have examined how academics encourage student engagement in online learning environments. They also investigated what teachers do and why they do it. The study that explored this problem on the LMS was not discovered by the researcher in a South African context. The intent of this study is to fill the gap in the literature. Participants were 116 academics from the faculties of A (76%) and B (24%). The question posed was: Which of the LMS tools were used effectively by the academics to foster students' engagement? To answer this question, embedded design was used in a mixed-method approach. Data were gathered using a survey questionnaire with both open-ended and closed-ended questions and interviews. Quantitative data were analyzed with the SPSS version 29 frequency distribution and percentage. Qualitative data were analyzed using Atlas.ti version 22. The results showed that 90.5% of the academics concurred that the learning activities on the LMS foster students to interact and engage. Instructional designers should support academics in the selections and the use of appropriate engagement tools on the LMS. The findings revealed that academics used LMS and third-party tools relating to the discussion forums, groups, Collaborate Ultra, Microsoft Teams chats, and WhatsApp to encourage interactivity in an online environment, as well as the development of authentic assessments in the LMS in this era of artificial intelligence.

**Keywords:** learning management system (LMS); digital tools; student engagement; higher education





## 1. Introduction

Online course management systems are widely used in higher education [1], and learning management systems (LMS) have served as the main platform for many universities to deliver e-learning [2]. Many universities choose to use LMSs as their primary e-learning delivery platform because they can provide a higher level of data continuity, stability, and privacy than the currently available free programs [2]. It is crucial that teachers base the design of online learning environments for their students on an implicit or explicit theory or framework of how they believe students learn [3]. Constructivist and connective theories, as well as student-centered learning models, were taken into consideration during the design and development of the LMS modules to make sure that students in this study engaged with the LMS tools. In this instance, the research demonstrates that, despite claims to the contrary, an LMS may not be able to improve teaching and learning by providing possibilities for social and constructive learning [2]. Even though higher education institutions in South Africa had reached a high level of digital maturity, it was still clear during the COVID-19 pandemic that the majority of academics used the LMS as the repository for sharing communication and teaching resources [4]. Barbetta [5] argued that the challenge of adopting LMS in higher education is to find innovative and effective active learning strategies that engage online students academically.

International studies have examined how academics encourage student engagement in online learning environments [6–11]. They also investigated what teachers do and why they

do it. The study that explored this problem on the LMS was not discovered by the researcher in a South African context. The intent of this study is to fill the gap in the literature. Further research on the ideal level of functionalities that an LMS can provide was suggested by Zanjani et al. [2]. This study examined this phenomenon in the context of higher education. Higher education institutions (HEIs) should support e-learning consciously because students' online engagement will increase significantly [6]. Complex tools and technologies are not necessary to encourage student engagement in the LMS [3]. According to Zanjani et al. [2], a LMS actually has the collaboration tools to offer opportunities for knowledge sharing, creating a community of learners, and fostering higher-order learning and critical thinking using discussion and collaborative learning. In this regard, Simelane-Mnisi and Mokgalaka-Fleischmann [4] made a case that the integration of digital LMS tools and resources into transformational pedagogies improves deep learning. To engage students in an authentic and real-world learning environment, it is essential to design learning activities that promote interaction, participation, and engagement [12]. In actuality, one of the key components of efficient LMS utilization is the LMS's interactive learning design [2,4]. According to Zanjani et al. [2], a user-friendly structure, avoiding too many tools and links, allowing privacy and anonymous posting, and more customizable student-centered features are LMS design elements that affect user engagement. Elements relating to affective/emotional engagement, social engagement, cognitive engagement, and affective/behavioral engagement should be taken into consideration while designing the LMS modules to promote effective learning [13–15].

The purpose of this study was to investigate the effectiveness of digital tools used by academics to foster students' engagement with regard to the LMS. To accomplish this, the survey questionnaire with closed-ended questions was used to determine the interactive learning design, development, and implementation of the LMS modules. Furthermore, this instrument was also used to establish the academics' attitudes towards the interactive learning design of the modules on the LMS. The open-ended semi-structured individual interview questions were applied to discover the LMS and other digital tools used by academics to promote student engagement in an online environment.

## 2. Related Literature

### 2.1. The Use of LMS during COVID-19

During the COVID-19 epidemic, the majority of HEIs were able to transition to online or remote learning for their teaching, support, and assessment methods [16]. HEIs were required to create and disseminate situational criteria to provide classes online and encourage students' participation [17]. Several institutions in Africa held discussions on online learning during the COVID-19 outbreak and were actively involved in making sure that their students had the best learning experience while experiencing the least amount of stress [18]. In the Republic of Trinidad and Tobago, the my-eLearning teaching platform was an essential tool for the continued support of students during the restrictions on face-to-face learning; even exam-related activities were carried out in this platform [18]. The majority of Malaysia's 20 public institutions were found to have responded by promoting or mandating online learning, whether using in-house e-learning platforms, live broadcasting on Facebook or YouTube, Lightboard Video Technology, or Zoom [19]. In Tanzania, Moodle and other LMSs were primarily developed and implemented [20].

### 2.2. LMS Digital Tool

The majority of digital tools require academics to design interesting activities and tasks for students, which are included in the LMS' key features [3]. Simelane-Mnisi and Mokgalaka-Fleischmann [4] stated that an LMS gives academics the freedom to customize it with dynamic digital technologies in order to promote effective teaching and learning. The digital tools used by the lecturers in the synchronous online activities to increase student engagement and attention were found to include assignment submission, personalized emails, commented slides, interactive videos, game-based learning applications, online

quizzes, and discussion forums in the study by Heilporn, Lakhal, and Bélisle [7] in the blended environment. These authors stated that the usage of various digital tools encourages undergraduate students' emotional and behavioral engagement. Crawford et al. [19] highlighted that web conferencing tools like Zoom, Webinar, and Panopto were used by Singaporean Polytechnic Universities and were linked to their LMS.

According to a report from a South African university, Chemistry teachers used the LMS's digital tools to support both synchronous and asynchronous learning. These tools fell into the categories of content, collaboration, communication, assessment, video conferencing, management, and survey [4]. The LMS server, folders, items, and files made up the content tools. A discussion forum, a journal for reflection, groups, a wiki, and blogs were among the collaborative tools. Announcements, chats (WhatsApp class group), email, and calendars were all used as communication tools. The LMS test, Microsoft Forms, Respondus, assignments, rubrics, and the plagiarism detector tool made up the assessment tools. For live teaching or webinars, the video conferencing platform LMS Collaborate Ultra was used. For monitoring and tracking students-at-risk, the management tools comprised reports, a grade center, and a retention center. The enterprise survey for student lecturer evaluation consisted of the survey tool.

### 2.3. Learning Design on Student Engagement

Engagement in online learning is defined as active participation in e-learning activities enabled by an e-learning platform [6]. The three key variables impacting students' academic progress in higher education have been recognized as student engagement, in-depth learning, and student satisfaction [7]. The amount of time and effort students intentionally put into their academic work and learning activities is known as student engagement [12]. Andrews and Fouche [13] argued that the ability of lecturers to design the order of activities, the resources needed, the timing, the establishment of structures and processes inside an LMS effectively, and the management of the learning pace improved student engagement in the LMS considerably. The study conducted by Zanjani et al. [2] about the LMS structure requisites that affected user engagement. It was discovered that a user-friendly structure, avoiding using too many tools and links, supporting privacy and anonymous posting, and more customized student-centered capabilities are LMS design elements that have an impact on user engagement.

It is crucial that academic, behavioral, cognitive, and affective/emotional variables are considered during the planning, design, and development phases of the online modules. The three independent but interconnected elements embracing the idea of student engagement on the LMS are behavioral, cognitive, and affective/emotional dimensions [15]. Finn and Zimmer [14] highlighted the four dimensions of academic, social, cognitive, and affective engagement as crucial components to foster student engagement. Academic engagement is described as the learning process, including the completion of assignments, paying attention, and participating in academic activities [14]. Behavioral engagement includes participation, interaction, collaboration, success in learning activities, performance, achievement, and skill development. Salas-Pilco, Yang, and Zhang [21]. Students who participate in online interface manipulation actively via behaviors like clicking, navigating, publishing, and scrolling frequently exhibit behavioral engagement in online learning Kennedy [3]. The degree to which students adhere to the rules of the class, such as arriving on time and communicating clearly with their classmates and the instructor, is a measure of their social engagement [14].

Cognitive engagement relates to students' goals, motivations for learning, willingness to put in the effort to understand, self-regulated learning, self-efficacy, perception of their abilities, capacity for reflection and critical thinking, a further effort to learn more, and the ability to grasp complex concepts. In this regard, Heilporn, Lakhal, and Bélisle [7] stated that the application of learning or metacognitive strategies by students, as well as their emotional commitment to mastering complicated learning, are required for the indications of cognitive engagement. This indicates that in online learning environments, students

who are more actively involved in the course material exhibit cognitive engagement [21]. Kennedy [3] supports Salas-Pilco, Yang, and Zhang [21] and further adds that cognitively engaged online students are driven to learn and demonstrate self-regulated learning.

Affective/emotional engagement refers to students' attitudes toward teachers, peers, and courses, as well as their appreciation of the material and the learning environment and their feelings of satisfaction and well-being [21]. These authors further highlighted that online learners who are emotionally engaged are those who exhibit and communicate positive attitudes while studying.

### 2.4. Theoretical Framework: ADDIE Model

Instructional design is at the core of effective learning and teaching because it is still very important to curriculum designers, instructional designers, learning experience designers, and academics. Most instructional designers use the ADDIE model as their framework [22,23]. The five (5) phases of the ADDIE model—Analysis, Design, Development, Implementation, and Evaluation—have flexible guidelines that assist instructional designers in creating successful support systems [23–25]. The ADDIE model was the theoretical framework that grounded this study. However, for the purpose of this paper, the Design, Development, and Implementation were the constructs applied to understand the effectiveness of LMS digital tools used by academics to foster student engagement.

During the design phase, the instructional designer focuses on selecting a course style, creating a suitable instructional approach, and creating an effective assessment technique for the subject [23,24]. It is worth indicating that the Design phase is often conducted after the Analysis phase, which led to the development of the storyboard and prototype in this study. The intention of the design phase was to ensure that constructive alignment was adhered to as indicated in the approved HEQSF curriculum. This implies that the learning outcomes, assessment criteria, content and activities, and learning and teaching approaches were included in designing the final product. The development of the storyboard and prototype was based on the curriculum documents, such as the module descriptor and study guide, which form part of this phase.

Frequently, instructional designers assert that the development phase is where the design phase's elements come to life [23,24]. In this study, during the Development phase, the instructional designer ensured that academics used the storyboard and prototype as the guiding principle in the development of interactive online modules on the LMS with various third-party tools such as discussion forums, groups, Collaborate Ultra, Microsoft Teams chats, and WhatsApp that encourage interactivity in an online environment. The LMS templates were used as they assisted with the scaffolding and chunking of the learning content and activities, which allowed problem-centered activity. Clear guidance was considered to allow students to know exactly what was required of them. LMS digital tools that foster student engagement, participation, and interaction were used in each learning unit. This was to afford a better user experience on the LMS.

The implementation phase deals with bringing a plan into action and comprises three main steps related to training, preparing students, and setting up the learning environment [23,24]. In this study, the prototype, which was designed and developed on the LMS, was used during the implementation phase. The academics were empowered with online facilitation skills. The academics have to evaluate the developed prototype before implementing it with the students. Necessary adjustments and refinements were applied. Student orientation on the use of the LMS digital tools was conducted before the implementation. Students automatically obtain access to all the modules they are registered for on the LMS; however, the academics had to open the learning materials for students to access. The researcher included the attitude construct to test the user satisfaction of the interactive online modules on the LMS.

## 3. Methods

The question posed in this study was: Which of the LMS tools were used effectively by the academics to foster students' engagement? Mixed-methods research with an embedded design was employed to address this question. The mixed method was applied because it incorporates the elements of qualitative and quantitative research in a single study, as emphasized by the pragmatic paradigm [26,27]. According to Cohen, Manion, and Morrison [26], the pragmatist theory, which recognizes and deals with the fact that the world is neither completely quantitative nor quantitative but is, rather, a mixed world, is the theoretical foundation for mixed-methods research. An embedded design refers to the integration of qualitative data into quantitative data or vice versa [27]. In this study, both quantitative and qualitative data were gathered at the same time using a survey questionnaire containing both open-ended and closed-ended questions and interviews. Separate analyses of the data were conducted, and quantitative data were analyzed with the SPSS version 29 frequency distribution. Saldaña's thematic method of analysis was used in qualitative data analysis [28]. Thematic analysis is the practice of locating patterns of meaning (themes) using codes [29].

## 4. Participants

Participants were chosen for the study using stratified purposive sampling [26]. The researcher selected academics from the population of the two faculties at the Study University of Technology in South Africa. Additionally, the researcher used purposive sampling to select a sample from these faculties. It may be inferred from this that the participants were 116 academics from the faculties of A (76%) and B (24%). The academics were chosen due to their participation in the online survey.

The result showed that based on the faculty, gender, highest qualification, and position, the majority (98.3%) of the academics' modules were active on LMS. Of these academics, 75.1% were from Faculty A. Less than three-quarters (64.7%) of the females' modules were active on LMS. It may be seen from Table 1 that slightly more than half (51.7%) of the academics had a doctorate or Doctor in Technology Degree. Less than three-quarters (63.8%) of the academics that had a lecturer position had modules on the LMS.

**Table 1.** Cross-tabulation of Participants' Demographic Data.

| | | Module on LMS | | |
| --- | --- | --- | --- | --- |
| | | **Yes** | **No** | **Total** |
| Faculty | A | 86 | 2 | 88 (75.9%) |
| | B | 28 | - | 28 (24.1%) |
| Total | | 114 (98.3%) | 2 (1.7%) | 116 (100%) |
| Gender | Female | 75 | 2 | 77 (66.4%) |
| | Male | 39 | - | 39 (33.6%) |
| Total | | 114 (98.3%) | 39 (33.6%) | 116 (100%) |
| Highest Qualification | Doctorate Degree or DTech | 60 | 2 | 62 (53.4%) |
| | Master's Degree or MTech | 45 | - | 45 (38.8%) |
| | Bachelor's Degree (Honors) or BTech | 9 | - | 9 (7.8%) |
| Total | | 114 (98.3%) | 2 (1.7%) | 116 (100%) |

**Table 1.** *Cont.*

| | | Module on LMS | | |
|---|---|---|---|---|
| | | **Yes** | **No** | **Total** |
| Position | Professor | 3 | 0 | 3 (2.6%) |
| | Associate Professor | 7 | 1 | 8 (6.9%) |
| | Head of Department | 5 | - | 5 (4.3%) |
| | Section Head of Department | 2 | 1 | 3 (2.6%) |
| | Senior Lecturer | 18 | - | 18 (15.5%) |
| | Lecturer | 74 | - | 74 (63.8%) |
| | Junior Lecturer | 4 | - | 4 (3.4% |
| | Laboratory Technician | 1 | - | 1 (0.9%) |
| Total | | 114 (98.3%) | 2 (1.7%) | 116 (100%) |

## 5. Instrument and Procedure

### 5.1. Survey Questionnaire

The survey questionnaire consisted of closed-ended questions pertaining to the quantitative elements of this study. Data on the academics' demographic data were elicited in Part A of the questionnaire. Data pertaining to the design, development, implementation, and attitude of LMS modules were included in Part B. The four factors of this instrument were (a) Design (5 items), (b) Development (3 items), (c) Implementation (3 Items), and (d) Attitude (2 Items). Part C consisted of data collected from 3 open-ended questions. The scale utilised the following ratings: 5 = Strongly Agree, 4 = Agree, 3 = Neutral, 2 = Disagree, and 1 = Strongly Agree

### 5.2. Open-Ended Questions

Three open-ended questions constituted Part C of the questionnaire. The questions asked were: (1) Identify the tools you used to promote interactive learning on myTUTor. (2) Which assessment tools did you use to measure students' understanding of the content? (3) Were the learning material and activities on myTUTor organized to promote the interaction, participation, and engagement between the students and lecturer? Yes or No? Elaborate.

### 5.3. Interview

The individual, semi-structured interviews used a similar question as the relevant open-ended question to confirm and triangulate the findings. During lockdown, the interviews were carried out using Skype. The teachers were then emailed the transcripts of the recorded interviews to ensure that the data had been recorded appropriately.

## 6. Results and Discussion

The results showed the internal consistency scores for the entire questionnaire, which consisted of 13 items. Cronbach's alpha [30] values were 0.90. The values for the items' alpha values ranged between 0.88 and 0.89. This indicated that the items had a comparatively best level of internal consistency. According to the reliability test, 0.70 and above is considered good. The best scores were found above 0.90, which is even more trustworthy than 80 [31].

The exploratory factor analysis was used to determine the validity of scores from the survey instrument. The results showed the values for Kaiser-Meyer-Olkin (KMO) and Bartlett's test of sphericity were found to be 0.880, and it was statistically significant ($p < 0.05$). Face validity was established because, according to the literature [32], KMO values between 0.7 and 1 indicated that the sampling was adequate.

An exploratory factor analysis was undertaken to establish the four-factor solution. The factors related to design (5 items), development (3 items), implementation (3 Items), and attitude (2 Items). In this study, a factor loading value of 0.5 was used as the cut-off point. Table 2 shows the principal component analysis with a varimax rotation of the survey questionnaire. The varimax rotation produced four variables, accounting for 72.7% of the total variation. It may be observed that during Development 1 (I created and built all content and components based on the design phase), it was cross-loaded and was removed. During Development 2 (I constructed, scaffolded, and chunked the learning content and activities based on the structure for each unit/chapter/topic) loaded with the design. Based on the four factors, the exploratory factor analysis was acceptable.

**Table 2.** Principal Component Analysis with Varimax Rotation of the Survey Questionnaire.

|  | **1** | **2** | **3** | **4** |
|---|---|---|---|---|
| Design 4 | 0.784 |  |  |  |
| Design 2 | 0.729 |  |  |  |
| Design 3 | 0.716 |  |  |  |
| Development 2 | 0.700 |  |  |  |
| Design 1 | 0.641 |  |  |  |
| Design 5 | 0.550 |  |  |  |
| Implementation 2 |  | 0.873 |  |  |
| Implementation 3 |  | 0.704 |  |  |
| Development 3 |  | 0.575 |  |  |
| Attitude 2 |  |  | 0.876 |  |
| Attitude 1 |  |  | 0.623 |  |
| Implementation 1 |  |  |  | 0.876 |
| Eigenvalues | 46.5 | 10.9 | 9.1 | 6.0 |
| Total Variance (%) | 25.9 | 18.9 | 15.4 | 12.4 |

It is crucial to indicate that the validity and reliability of all the scores were guaranteed. This indicates that the findings were disclosed with the confidence that they were genuine and reliable. All (116) participants completed the questionnaire. Table 3 shows the results with respect to the 13 items.

**Table 3.** Results from Survey Questionnaire.

|  | **Strongly Agree** | **Agree** | **Neutral** | **Disagree** | **Strongly Disagree** |
|---|---|---|---|---|---|
| | | | Design | | |
| D 1 | 65 (56.0) | 38 (32.8) | 10 (8.6) | 1 (0.9) | 2 (1.7) |
| D 2 | 65 (56.0) | 40 (34.5) | 9 (7.8) | 2 (1.7) | - |
| D 3 | 51 (44.0) | 49 (42.2) | 15 (12.9) | 1 (0.9) | - |
| D 4 | 34 (29.3) | 45 (38.8 | 29 (25.0) | 7 (6.0) | 1 (0.9) |
| D 5 | 85 (73.3) | 25 (21.6) | 4 (3.4) | 2 (1.7) | - |
| | | | Development | | |
| Dev 1 | 52 (44.8) | 47 (40.5) | 10 (8.6 | 6 (5.2) | 1 (0.9) |
| Dev 2 | 42 (36.2) | 46 (39.7) | 20 (17.2) | 7 (6.0) | 1 (0.9) |
| Dev 3 | 74 (63.8) | 34 (29.3) | 7 (6.0) | - | 1 (0.9) |

**Table 3.** *Cont.*

|  | Strongly Agree | Agree | Neutral | Disagree | Strongly Disagree |
|---|---|---|---|---|---|
| Implementation | | | | | |
| I 1 | 33 (28.4) | 46 (39.7) | 26 (22.4) | 7 (6.0) | 4 (3.4) |
| I 2 | 78 (67.2) | 27 (23.3) | 4 (3.4) | 6 (5.2) | 1 (0.9) |
| I 3 | 67 (57.8) | 42 (35.3) | 7 (6.0) | 1 (0.9) | - |
| Attitude | | | | | |
| A 1 | 66 (56.9) | 39 (33.6) | 10 (8.6) | 1 (0.9) | - |
| A 2 | 42 (36.2) | 44 (37.9) | 21 (18.1) | 6 (5.2) | 3 (2.6) |

In terms of the five items relating to the design, it may be observed from the table that in Design 5, the majority (94.9%) of the academics agreed that they created module contents or materials, such as PowerPoint presentations, notes, videos, and audio, for example. It may be argued that academics were able to use various multimedia technologies to create module content that accommodated various learner preferences to ensure that the learning outcomes were achieved. This finding is supported by Chen et al. [33], who indicated that rich visual and auditory information from multimedia-based learning tools engage many sensory systems, and using video over text resources enhances outcomes. The results also showed that in Design 2, the majority (90.5%) of the academics agreed that they used the module descriptor or study guide to establish outcomes and assessment criteria for each learning unit/chapter/topic. It was imperative that during the design phase, constructive alignment be emphasized so as not to deviate when developing the LMS module. Razeed and Werkhoven [34] argued that, based on the course's overall objectives, each topic should have its own set of learning outcomes. Simelane-Mnisi [35] emphasized that constructive alignment with the authorized curriculum design should be adhered to in an online module.

Regarding Development 3, it was found that the majority (93.1%) of the academics agreed that they made the module available to students on selected media and tools of delivery. It was critical that academics should give access to students to the developed online module with the use of various LMS tools. Razali et al. [36] mentioned that diverse learning strategies that give students a more accessible and flexible approach to learning using the LMS are a necessity. The results revealed that in Development 1, most (85.3%) of the academics agreed that they constructed, scaffolded, and chunked the learning content and activities based on the structure for each unit/chapter/topic. In this case, Sinnayah, Salcedo, and Rekhari [37], in their study, academics were provided with a variety of H5P templates to pick from and were appropriate for their chosen topic; this integration makes it simple to produce new content within the online teaching environment.

In terms of Implementation 2, it was found that the majority (90.5%) of the academics agreed that the learning activities on the LMS motivated students to interact and engage. It was imperative that academics develop online materials that encouraged students to be actively engaged with online tasks that instilled creativity by developing creative ideas and problem-solving [34]. The results revealed that in Implementation 3, the majority (93.1%) of the academics agreed that they presented online lectures/online classes/live lecturers' webinars on Collaborate/Zoom/Google Meets. It was also found that video conferencing tools supported academics to deliver the lesson online. In this case, Goshtasbpour et al. [38] indicated that the learning support was increased while the course was being delivered using providing webinars.

Regarding Attitude 1, most (89.9%) of the academics agreed that they liked the idea of using LMSs during the lockdown and should continue using them even when they were back on campus. The deployment of technologies worldwide, such as the LMS and the application of the tools, have helped the majority of institutions greatly to reach the maturity

level regarding the use of LMS [36]. The result revealed that almost three-quarters (74.1%) of the academics agreed that the LMS activities were effective in promoting active learning during the learning process. It may be argued that to encourage student engagement in the LMS, various tools should be designed and developed to promote academic, behavioral, cognitive, and affective/emotional attributes [14,15]. For the purpose of promoting learning in the LMS and creating an engaging, accessible environment, it is crucial that academics select relevant resources and tools.

## 7. Qualitative Findings

Three questions were used to gather data from closed-ended questions and semi-structured interviews. There were about 21 academics from each department in the Faculty of A and B who took part in the interviews. The trustworthiness of this study was assured based on credibility, dependability, transferability, and confirmability to assess the quality of qualitative research [22]. Atlas.ti version 22 was used to analyze the qualitative data using a thematic approach and discover significant patterns (themes) in the codes [25]. One primary document was used to create 63 codes. The system generated 26 quotations. The codes were grouped into three networks that dealt with LMS digital tools, assessment tools, and engagement.

### 7.1. LMS Digital Tools

The theme of the LMS digital tools emerged from the question: Identify the tools you used to promote interactive learning on myTUTor. The findings showed that various LMS and third-party digital tools were used by academics during the design and development of the modules in the LMS. Figure 1 illustrates the LMS digital tools used.

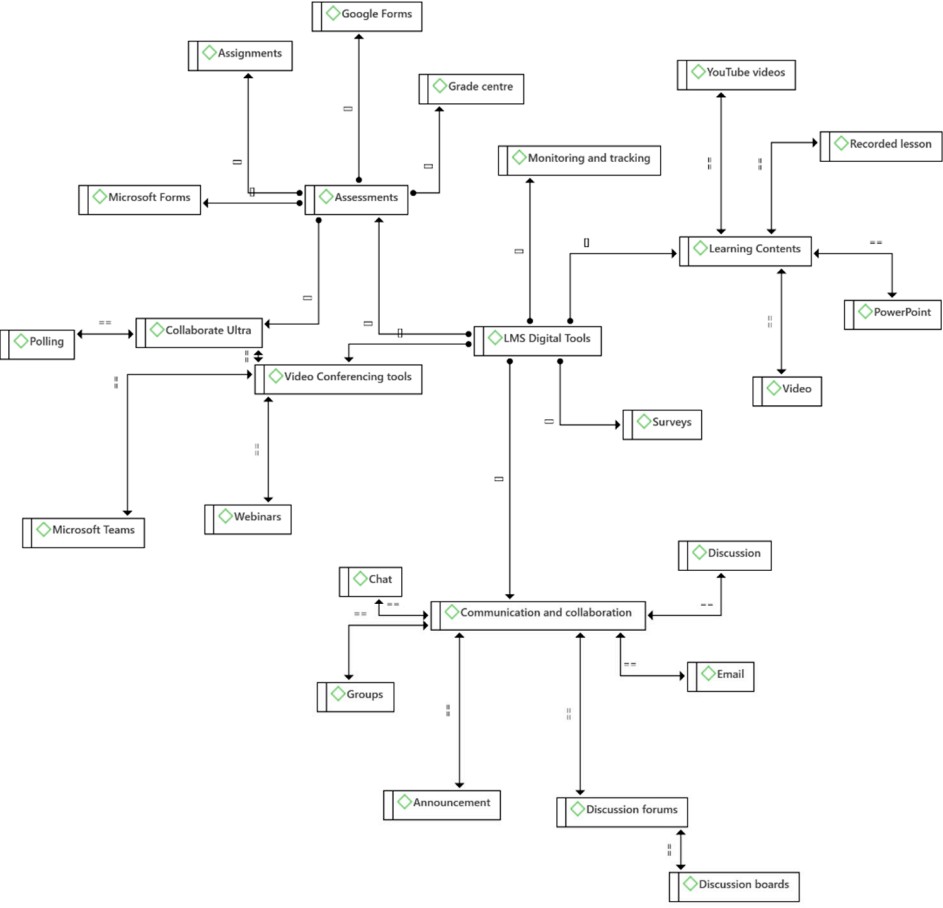

**Figure 1.** The network or conceptualization of LMS digital tools. (The symbol [] means the category connection is part of the theme and = = means the code connection is associated with the category.)

The aim of using the LMS digital tool was to emphasize student engagement, interaction, and participation. After looking closely at Figure 1, it may be argued that these tools are related to categories such as content, communication and collaboration, assessment, video conferencing, monitoring, and tracking, as well as surveys. The categorization of tools was also found in the study conducted by Simelane-Mnisi and Mokgalaka-Fleischmann [4]. The researcher may argue that academics designed and developed their modules with the aim of encouraging interactive learning in an online environment. This finding is supported by Simelane-Mnisi [35], who revealed that one of the most effective methods to encourage students to constantly desire to learn is via interactive learning.

It was found that academics used the content tools relating to learning content, recorded lessons, voice-over PowerPoint presentations, YouTube videos, and videos recorded by the academics. This finding was supported by 94.9% of the academics in this study who created the module contents with the aid of multimedia content to support the online implementation of the flipped learning strategy [12]. The LMS was used to make learning resources and materials available to students to access asynchronously or synchronously.

The finding revealed that the communication and collaboration tools utilized by the academics involved announcements, emails, chats, discussion forums, or discussion boards and groups. These communication tools have the potential to encourage behavioral engagement among the students as learning activities developed with them could cultivate collaboration, communication, and interaction [21]. In addition, these communications interactive learning tools have the potential to encourage active learning among students [39]. These tools also supported social engagement in the LMS [14].

It was discovered that the assessment tools applied by the academics comprised grade centers, assignments, assessments, and Microsoft Forms. The application of these assessment tools supports the development of online assessment activities that encourage a student-centered approach [34]. Peer or self-evaluation systems and relevant feedback are some of the tools that can be used to increase students' engagement [6]. To ensure student engagement, assessment tools could be applied in the LMS to encourage students to be more engaged with the topic and exhibit cognitive engagement [21].

The findings revealed that the video conferencing tools employed by the academics involved Collaborate Ultra and Microsoft Teams, where webinars were conducted. This finding is reinforced by 93.1% of the academics in this study, who revealed that they used these tools to facilitate learning. The literature revealed that video conferencing tools were incorporated into educational environments, especially on the LMS and other eLearning platforms [12]. The study conducted by Ndovela, Risinamhodzi, and Matobobo [40] in South Africa revealed the challenge of learning and teaching with Microsoft Teams was the absence of direct student engagement. However, the study on increasing student participation and involvement in live Microsoft Teams classes by Simelane-Mnisi and Mangavana [12] explored how such technology helped with student engagement. Monitoring and tracking tools were also utilized as survey tools. It was found that LMS was used for tracking students' access to embedded H5P activities [36]. These authors also indicated that they used surveys to measure students' perspectives on the effectiveness of the various activities.

### 7.2. Assessment Tools

The theme assessment tools were generated from the question Which assessment tools did you use to measure students' understanding of content? Educational assessment entails gathering and analyzing information about student knowledge, skills, and values in learning, teacher effectiveness, classroom and online management, and institutional procedures, among other topics [41,42]. It was found in this study that various assessment tools on the LMS, including third-party assessing tools, assessment types, instruments, and methods, were applied. The LMS assessment tools that were utilized included assignments, testing/quizzes, and polling tools. The third-party assessment tools that were utilized linked to the LMS consisted of Google Forms and Microsoft Forms. It was found in

this study that the LMS assessment tools were applied with different assessment types, such as the assessment as, for and of learning, formative assessment, and formative and summative assessment [42]. Several studies that investigated the effective use of LMS applied formative and summative assessment [34,37]. Assessments are conducted to support learning, enhance learning, certify students, and determine prior knowledge [41].

The findings revealed that academics used different assessment methods to conduct online assessments on the LMS. The methods included online tests, online examinations, tutorials, assignments, and polling with Collaborate Ultra. It was found that the academics applied different assessment instruments that required students to think deeply and critically to answer the questions. The form of assessment tasks given to the learner is referred to as the assessment instrument [42,43]. The assessment instruments that were utilized were open books, open-ended questions, case studies, essays, and orals. The researcher may argue that these assessment instruments are closely related to authentic assessment. Juanda [44] opined that authentic assessment helps students acquire a variety of graduate attributes by exposing them to "real world" situations. It was discovered that assessment instruments that test students' knowledge and understanding and provide immediate feedback were also applied. These assessment instruments are related to polling, quizzes, and self-tests. The questions that were used were multiple choice, matching, true/false, and crosswords. Rao and Banerjee [41] argued that these assessment instruments complement the increasingly popular computerized or online assessment formats well and have distinct, objective solutions. These assessment instruments are useful for assessing the student's overall comprehension of a subject as well as their ability to create and innovate [45].

It was discovered that the academics made use of the Grade book to capture the scores and view the progress of the students via various assessments conducted online. Sinnayah, Salcedo, and Rekhari [37] revealed that the teaching staff in their study had the ability to see the student progress and connect learner analytics to grade book systems, which was the key benefit of the LTI connection. It is critical in the LMS to enhance the content with activities, such as self-assessment and formative assessment, for students to do self-study and test their knowledge and understanding as they study online. Furthermore, students should be able to monitor their progress on the LMS with the use of grade books.

*7.3. Engagement*

The theme of engagement emerged from the question: Were the learning material and activities on myTUTor organized to promote interaction, participation, and engagement between the students and lecturer? Yes or No? Elaborate. HEIs have identified student engagement, in-depth learning, and student satisfaction as indicators of students' academic achievement [6]. The findings revealed that academics had mixed feelings about this question. Other academics indicated that, indeed, the learning materials they developed on the LMS promoted student engagement, whereas others felt otherwise.

In terms of the academics who indicated that interaction, participation, and engagement were promoted in their online modules, it was discovered that academics used LMS and third-party tools relating to the discussion forums, groups, Collaborate Ultra, Microsoft Teams chats, and WhatsApp to encourage interactivity in an online environment. This finding is supported by 90.5% of the academics in this study who created learning activities on the LMS to encourage student engagement in the LMS.

It was found that academics encouraged participation in various ways, which kept students enthusiastic. Academics said A8: "Yes. Most students enjoyed the sessions and were participating with great enthusiasm." A9 mentioned, "Yes. In all module units, there are a variety of activities or course tools that were integrated within the module. Our module was the pilot study for the faculty and received compliments on the originality and its practicality by both the staff and students". A28 indicated, "Yes. I included interesting bits of information to keep students interested, as well as polls and external quizzes. I used academic check-in and wrap-up activities that proved to be quite useful, too. The sessions were flexible, and students felt free to ask questions at any time during the sessions".

Tavakol and Dennick [31] revealed that more than 78% of students noted that Canvas LMS offered them options for engagement.

It was found that students participated in the discussion forum during the implementation. A19 responded, "Yes. Discussions boards were opened on important topics". A12 mentioned, "Yes, the discussion tool was often used to encourage participation and engagement." A3 indicated, "the discussion forums were interactive." This sentiment corroborates what Razeed and Werkhoven [34] has reported, namely that 68% of the students responded positively that the canvas activities allowed them to engage with other students via discussion forums.

Collaborate Ultra and Microsoft Teams were used to conduct live online classes or webinars, and group activities were conducted to encourage collaboration. A5 indicated, "I found that students do participate on webinars." A19 intimated, "During the online classes, students were divided into breakaway groups, which aided interaction." A14 affirmed, "Yes: Students were referred to activities to do before classes. Students participated in breakout rooms in smaller groups and within the online class, activities or scenarios were provided to encourage participation." Simelane-Mnisi and Mangavana [12] reported that Microsoft Teams breakout rooms were used to divide the class into groups, and students were engaged in the jigsaw activity. Breakout rooms in an online setting, according to Amelia and Yosintha [46], foster collaborative learning, creativity, and student involvement. It was also found that students were given the opportunity to present on Collaborate Ultra. A20 shared, "Students gave feedback on Collaborate Ultra using PowerPoint presentations, and they engaged in discussions. I could track participation better over a period of time than in a face-to-face class situation." Some of the academics experienced low attendance and participation during the live classes. A1 said, "Yes, I tried, but found low students' participation and involvement in webinars."

It was found the online group activities were conducted on the LMS. A16 said, "I created small groups online because bigger groups did not work for mathematics." A23 indicated, "I gave students assessments to work in groups, and I guided them all the way to the completion of the assessment." A17 mentioned, "I find dividing students into groups helpful. I then "visit" each group during the online workshop. I also encourage students to use the "chat" option. Senior students seem to engage more easily as they are used to group work".

It was found that other academics created WhatsApp groups and linked them on the LMS for better management of online modules. A5 noted, "Students are active on WhatsApp. The current situation actually resulted in more interactive communication." A16 indicated, "We also had a WhatsApp group that worked extremely well."

It was also found that other academics did not develop activities that required students to engage with the online materials; this was observed regarding the lack of participation from the students with online materials. A6 commented, "No, not really. Time was a problem to prepare for proper interaction, participation, and engagement; most students did not attend the online sessions." A4 indicated that "Students did not participate. It is maybe a lack of not knowing how to go about it on the students' side. Some excelled, others did nothing". A11 shared, "Students mostly do not engage with academics during online classes."

## 8. Conclusions

In this study, we saw how academics took advantage of and applied LMS digital tools to encourage student engagement because LMS provided a higher level of data continuity and stability. Academics in this study created their LMS modules while they were cognizant of how students learn. Various theories and methods of learning were thought through well in the design and development of online modules. This supported the selection and the appropriate use of the digital tools that fostered engagement. The learning activities that were designed by the academics on the user-friendly LMS structure encouraged active learning amongst the students. The significant aspects of student engagement include

academic, behavioral, cognitive, and affective/emotional aspects. It may be seen in this study that the LMS engagement tools that were used by academics included discussion forums, groups, Collaborate Ultra, Microsoft Teams chats, and WhatsApp, and these tools encouraged social engagement online.

## 9. Recommendations

Academics should be empowered regarding the selection and use of appropriate engagement tools as well as the development of authentic assessments in the LMS in this era of artificial intelligence. It is recommended that academics are provided with institutional support to design and develop their online modules to apply LMS digital tools that enhance engagement among the students. It is crucial that academics design and develop their modules as it is a skill that academics should exhibit in the 21st century, and learning experience designers or instructional designers should provide them with guidance and support instead of relying on them for the development of the online modules in the LMS. Active learning in the online modules should be fostered in the LMS. Further studies could be conducted on students' satisfaction and engagement with the use of the LMS. In addition, further studies could be conducted on the same phenomenon in different contexts.

**Funding:** This research was funded by [National Research Funding (NRF) Thuthuka Grant] grant number [138262].

**Institutional Review Board Statement:** The study was conducted in accordance with the Declaration of Sibongile Simelane-Mnisi, and approved by the Tshwane University of Technology Ethics Committee) of NAME OF INSTITUTE (REC2020/11/014, 2020/10/ and renewed 2023/07).

**Informed Consent Statement:** Informed consent was obtained from all subjects involved in the study.

**Data Availability Statement:** The data are not publicly available due to privacy or ethical restrictions.

**Acknowledgments:** I acknowledge the financial support from National Research Funding (NRF) Thuthuka Grant to conduct this study.

**Conflicts of Interest:** The author declares that she is the instructional designer at the study university and is responsible for the Faculty where the data was collected.

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
