# Peer review of "Effectiveness of LMS Digital Tools Used by the Academics to Foster Students’ Engagement"

_education, doi:10.3390/educsci13100980_

Round 1
Reviewer 1 Report
The Introduction section provides a comprehensive introduction to the topic of LMS in higher education and the research's purpose.The article addresses various facets of Learning Management Systems (LMS) in higher education, particularly during the COVID-19 pandemic, and how they impact student engagement. The article underscores the importance and efficacy of LMS in higher education, especially during times of crisis, and how properly designed LMS courses can significantly influence and enhance student engagement. The research question is clearly stated, making it easy for the reader to understand the focus of the study.The stratified purposive sampling technique is mentioned, which gives insight into how participants were chosen and ensures a diverse representation. Error in the given numbers: A (79.5%) and B (24.1%) - gives a sum over 100 (line 222). The section titled "Instrument and Procedure" breaks down the tools and techniques used in the study. The "Results and Discussion" section is an essential part of a research study as it interprets the findings and relates them to the broader context. The section provides a thorough review of the results, breaking them down by various aspects like Design, Development, Implementation, and Attitude. It provides a comprehensive look into how academics use digital tools within an LMS to promote student engagement in online environments.
Author Response
|
Comments 1: [Error in the given numbers: A (79.5%) and B (24.1%) - gives a sum over 100 (line 222)
|
|
Response 1: [Number given is corrected as follows, A (76%) and B (24%) gives a sum of100 in line10 and 221] Thank you for pointing this out. I agree with this comment. Therefore, I have….[included the correct percentage in page 1, paragraph 1, and line 10 as well as on page 5, paragraph 3, line 221.] “[updated text in the manuscript]” |

Reviewer 2 Report
Thank you for allowing me to evaluate this article. The issues to be corrected are as follows:
The summary should be corrected, as there are errors of expression in lines 20-21.
The versions of the software used should be added
Author Response
|
Comments 1: [The summary should be corrected, as there are errors of expression in lines 20-21.)
|
|
Response 1: Errors of expression in lines 20-21 is corrected] Thank you for pointing this out. I agree with this comment. Therefore, I have….[corrected the summary’s error of expression in page 1, paragraph 1, and line 20-12]. “[updated text in the manuscript]” |
|
Comments 2: [The versions of the software used should be added.] |
|
Response 2: Agree. I have, accordingly, added the versions of the software used.to emphasize this point. [SPSS version 29 is included as suggested in line 14 and 212. Atlas.ti version 22 is included as suggested in line 15 and 33..] “[updated text in the manuscript]” |
